# Diversity of visual inputs to Kenyon cells of the *Drosophila* mushroom body

Ishani Ganguly [1,2,3,7], Emily L. Heckman [4,7], Ashok Litwin-Kumar [1,2,3], E. Josephine Clowney [4,5] ✉ & Rudy Behnia [1,3,6] ✉

The arthropod mushroom body is well-studied as an expansion layer representing olfactory stimuli and linking them to contingent events. However, 8% of mushroom body Kenyon cells in *Drosophila melanogaster* receive predominantly visual input, and their function remains unclear. Here, we identify inputs to visual Kenyon cells using the FlyWire adult whole-brain connectome. Input repertoires are similar across hemispheres and connectomes with certain inputs highly overrepresented. Many visual neurons presynaptic to Kenyon cells have large receptive fields, while interneuron inputs receive spatially restricted signals that may be tuned to specific visual features. Individual visual Kenyon cells randomly sample sparse inputs from combinations of visual channels, including multiple optic lobe neuropils. These connectivity patterns suggest that visual coding in the mushroom body, like olfactory coding, is sparse, distributed, and combinatorial. However, the specific input repertoire to the smaller population of visual Kenyon cells suggests a constrained encoding of visual stimuli.

In expansion layer circuits, large populations of neurons receive combinatorial inputs from smaller populations of neurons carrying diverse sensory signals. Prominent examples include the lophotrochozoan parallel lobe system and chordate cerebellum, hippocampus, pallium, and cortex. A well-studied example is the mushroom body of *Drosophila melanogaster*, where ~2000 intrinsic neurons, called Kenyon cells, link sensory representations to proximate events via dopamine-mediated synaptic plasticity[1–6]. The mushroom body is thought to have evolved as an olfactory structure, and in *Drosophila*, more than 90% of Kenyon cells receive predominantly olfactory input[7–9]. However, 8% of Kenyon cells in *Drosophila* receive mainly visual inputs, and radical expansions of the visual mushroom body have occurred independently in several arthropod clades[7,9].

Flies have two subsets of visual Kenyon cells. KCγ-d cells (KCγ-d's) are the first Kenyon cells to be generated in development[10–12]. They receive olfactory input in the larval mushroom body but re-wire during metamorphosis to innervate the ventral accessory calyx (vACA)[7,10,13,14].

In adults, they have been shown to receive direct input from visual projection neurons (VPNs) from the medulla through VPN-MB1 and VPN-MB2[15] and the accessory medulla through aMe12[7,16]. KCαβ-p cells (KCαβ-p's) are born during pupal stages, before olfactory αβ cells, and innervate the dorsal accessory calyx (dACA)[10]. They have been shown to receive indirect visual input through local visual interneurons (LVINs) processing signals from the lobula[17,18]. These experimental observations were augmented by a description of the repertoire of visual interneurons providing inputs to the mushroom body in the hemibrain connectome dataset, which highlighted the previously unexpected extent of visual projections to this area[7].

The visual system of the fruit fly is composed of four main neuropils: the lamina, the medulla, the lobula and the lobula plate. The accessory medulla is a lesser-known small optic lobe neuropil, tucked between the medulla and the lobula, where the extraocular photoreceptors of the Hofbauer–Buchner eyelet send their axons. aMe neurons receive direct input from retinal photoreceptor cells as well as

[1]Department of Neuroscience, Columbia University, New York, NY, USA. [2]Center for Theoretical Neuroscience, Columbia University, New York, NY, USA. [3]Zuckerman Institute, Columbia University, New York, NY, USA. [4]Department of Molecular, Cellular, and Developmental Biology, University of Michigan, Ann Arbor, MI, USA. [5]Michigan Neuroscience Institute, University of Michigan, Ann Arbor, MI, USA. [6]Kavli Institute for Brain Science, Columbia University, New York, NY, USA. [7]These authors contributed equally: Ishani Ganguly, Emily L. Heckman. ✉e-mail: jclowney@umich.edu; rb3161@columbia.edu

from eyelet photoreceptors[16,19]. Clock neurons also innervate the accessory medulla[20–23]. The role of the accessory medulla has not been investigated in *Drosophila*, but in other insect species, it has been implicated in entrainment of the circadian clock[24]. While analyses of the hemibrain connectome were able to identify visual interneurons providing input to the mushroom body and found that VPNs from the medulla, lobula, and aMe innervate the calyx, the identity of these VPNs could not be determined due to the lack of comprehensive reconstruction of the optic lobe neuropils[7,25].

In adult *Drosophila melanogaster*, olfactory Kenyon cells have between 3 and 10 large, claw-shaped dendrites that each receive input from a single, multisynaptic "bouton" deriving from cholinergic olfactory projection neurons in the antennal lobe[26–31]. Individual Kenyon cells only spike when multiple of their inputs are active. They thus form a sparse, combinatorial code for chemical features of odor space that improves stimulus discriminability as compared to the antennal lobe representation[32–34]. In contrast to the well-studied olfactory representation, it is still unclear what kind of visual information is conveyed to the MB, how visual information is encoded in Kenyon cells, and what this information is used for. Whereas blue versus green color learning[1,15,35,36] and context generalization[37] have been shown to specifically require the mushroom body, multiple visual learning paradigms, such as place learning[38] and pattern recognition[39,40], have been shown to rely on the central complex (CX). In terms of multisensory learning, which has been established in the context of memory enhancement[41,42] and transfer between olfaction and vision[41], the association between olfactory and visual information has been suggested to rely on the mushroom body[43]. The mushroom body has been shown to be required for learning of color associations in honeybees, learning of spatial associations in cockroaches, and navigation in ants[44–46].

Here, we use the FlyWire adult whole brain electron microscopy connectome to describe the repertoires of inputs to individual visual Kenyon cells in *Drosophila melanogaster*[8,47–49]. Consistent with previous reports, direct VPNs dominate input to KCγ-d cells, while indirect visual input via LVINs dominates input to KCαβ-p cells[7,15,17]. We focus our analysis on KCγ-d's and find that individual Kenyon cells can receive inputs from mixtures of presynaptic partners, including from both VPNs and LVINs and from multiple optic lobe neuropils. Many KC-projecting VPNs receive input from large swathes of visual space, while some KC-projecting LVINs receive more spatially restricted signals across visual space, a wiring logic that would provide Kenyon cells with both information about full-field conditions of the environment and translational invariance for limited number of specific visual features. Previous work found that aMe12 accessory medulla neurons provide input to visual Kenyon cells, and we find that inputs from this small neuropil are prominent overall[7,16], providing a potential link between the circadian system and the MB. We find that like olfactory Kenyon cells, visual Kenyon cells receive sparse inputs, from 1–7 neurons each. Finally, we compare the input patterns to individual Kenyon cells between hemispheres and to a random model; input patterns vary across hemispheres in this brain and are consistent with a random sampling of available inputs, but certain inputs are highly overrepresented compared to others in both hemispheres. These connectivity patterns suggest that, on the one hand, visual coding in the mushroom body, like olfactory coding, is sparse, distributed, and combinatorial. On the other hand, the expansion coding properties appear different, as a specific repertoire of visual input types project onto a relatively small number of visual Kenyon cells.

## Results

### Visual projection neurons dominate input to the ventral accessory calyx, and local visual interneurons to the dorsal accessory calyx

Sensory information is conveyed to the dendrites of Kenyon cells in the mushroom body calyces. The main calyx contains the dendrites of olfactory Kenyon cells, representing 90% of the total population. Previous connectomics analyses of the hemibrain dataset revealed that other sensory streams make connections in accessory calyces[7]. Visual inputs segregate into two streams of information that target KCγ-d cells (KCγ-d's) in the ventral accessory calyx and KCαβ-p cells (KCαβ-p's) in the dorsal accessory calyx (Fig. 1A, B). Each population receives both direct and indirect inputs (through LVINs) from VPNs (Fig. 1A). In the hemibrain dataset, the proportion of direct vs. indirect inputs was shown to be different in the two accessory calyces: The ventral pathway is enriched in direct VPNs whereas the dorsal pathway is enriched in indirect connections[7].

In the FlyWire reconstruction, we observe 147 and 148 KCγ-d cells on the left and right respectively; and 67 versus 61 KCαβ-p's. As in the hemibrain dataset, VPNs make up the majority of visual input onto KCγ-d's (49 of 74 presynaptic partners, 75.8% of input synapses), while LVINs constitute about a third of the inputs (Fig. 1C, D). Another prominent input to the KCγ-d population is the large inhibitory neuron APL, which makes on average 16 and 10 synapses per KCγ-d in the left and right hemispheres respectively (comparable to the average number of synapses APL makes onto each KCγ-m). In contrast to a previous observation that KCγ-d's do not have clawed dendrites[7], we observe KCγ-d's that make bouton-claw and *en passant* synapses with incoming VPNs and LVINs (Supplementary Fig. 1A-C). To determine whether VPNs and LVINs differ in preferred synapse type onto KCγ-d's, we used synapse locations to roughly classify synapses between VPNs or LVINs and their KCγ-d partners as either *en passant* or bouton-claw (see "Methods"). We find that VPNs and LVINs make roughly equivalent proportions of bouton-claw and *en passant* synapses onto KCγ-d's, but that both classes of visual input make more bouton-claw synapses overall (Supplementary Fig. 1D).

For KCαβ-p's, LVINs dominate the information flow (62.5% of presynaptic partners/84% of synapses; Fig. 1F, G). Few inputs are shared between KCγ-d's and KCαβ-p's (Fig. 1I). A ranking of VPNs and LVINs in terms of total synapse numbers made onto KCγ-d's shows that while VPNs are the main inputs, several LVINs are among the highest connected neurons (Fig. 1E). In the dorsal pathway to KCαβ-p's, LVINs constitute the bulk of the inputs but several VPNs are intermingled among them (Fig. 1H). As LVINs are potential nodes for multisensory integration, receiving not only visual information through indirect VPNs but also information from other sensory modalities[7], the dorsal and ventral pathways likely contribute differently to visual vs. multimodal mushroom body functions. While we expect the indirect and potentially multimodal inputs to KCαβ-p's will be very interesting for future study, we elected here to focus subsequent analyses on the more numerous KCγ-d's, as their receipt of strong, direct VPN inputs allows us to infer their visual tuning.

### Direct visual inputs to the ventral accessory calyx derive from three optic lobe neuropils

We identified 49 VPNs that form connections of at least 5 synapses onto KCγ-d's in the left hemisphere of the FlyWire dataset. 18 of these derive from the medulla (44% of synapses), 18 from the lobula (26.4% of synapses) and 13 from the accessory medulla (29.5% of synapses; Fig. 2A–C). Like olfactory projection neurons, all direct VPNs are predicted to be cholinergic and, therefore, excitatory. None of the VPNs connecting to the vACA have assigned functions. In order to classify them further, we used both anatomy and connectivity to cluster these VPNs into putative cell types (Fig. 2D, see "Methods"). Among the 18 medulla VPNs, most are singlets: They do not cluster with any other MB-projecting visual neurons and appear only once in each hemisphere. The one exception is a cluster comprising three neurons (ME.PLP.36, ME.PLP.42, and ME.1557). Like medulla VPNs, most of the 18 lobula VPNs are singlets, though LO.SCL.7 and LO.SLP.12 forms a cluster. In contrast, aMe VPNs providing input to the left vACA can be classified into just four cell types aMe12, aMe20 or aMe26_L and

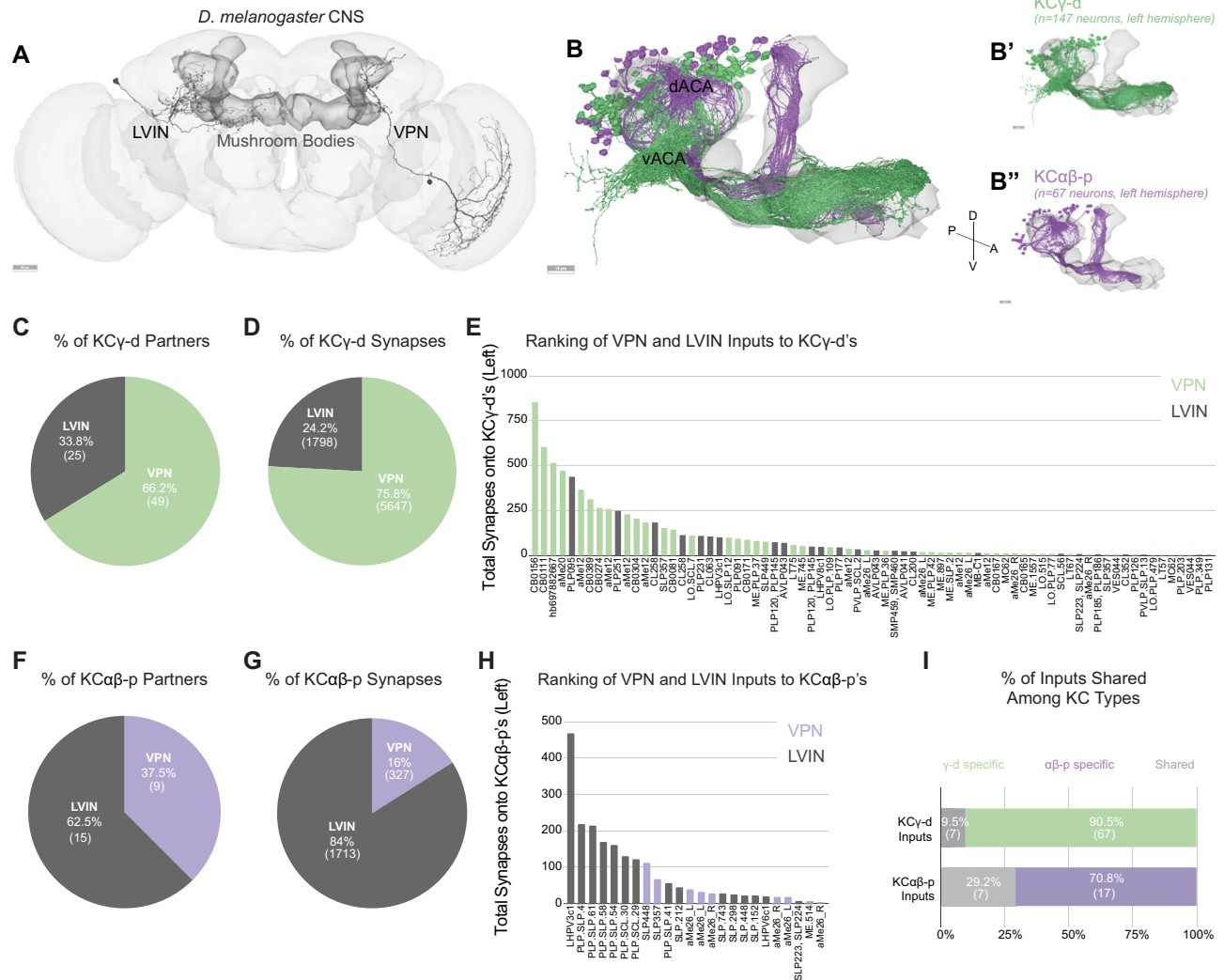

**Fig. 1 | Direct visual inputs are highly represented in the ventral accessory calyx. A** Schematic of the *D. melanogaster* CNS with the mushroom bodies highlighted in dark gray. Representative examples of an LVIN (PLP095) and VPN (CB0156). Anterior view. Scale bar, 30 μm. **B** Mushroom body volume with visual Kenyon cell subtypes overlaid. dACA, dorsal accessory calyx, vACA, ventral accessory calyx. (**B'**) KCγ-d, green receives input in the vACA. (**B"**) KCαβ-p, purple, receives input in the dACA. Scale bar, 15 μm. **C** Proportion of KCγ-d visual inputs that are either VPNs (green) or LVINs (gray). **D** Proportion of visual synapses onto KCγ-d that are either from VPNs or LVINs. **E** Ranking all VPN and LVIN inputs to KCγ-d by their total synaptic contribution to KCγ-d's. **F** Proportion of KCαβ-p visual inputs that are either VPNs (purple) or LVINs (gray). **G** Proportion of visual synapses

onto KCαβ-p that are either from VPNs or LVINs. **H** Ranking all VPN and LVIN inputs to KCαβ-p by their total synaptic contribution to KCαβ-p's. **I** Proportion of direct visual inputs and LVINs that are either specific to or shared among KCγ-d and KCαβ-p. All data in panels B-I examine connectivity to KCs in the left hemisphere, using a ≥ 5 synapse threshold. Source data are provided as a Source Data file. Figure 1A-B" neuron skeletons and neuropil volumes adapted with permission under CC BY-NC 4.0 license (https://creativecommons.org/licenses/by-nc/4.0/) from Dorkenwald et al. Neuronal wiring diagram of an adult brain. bioRxiv 2023.06.27.546656 (2023) https://doi.org/10.1101/2023.06.27.546656 and Schlegel et al. Whole-brain annotation and multi-connectome cell typing quantifies circuit stereotypy in *Drosophila*. bioRxiv 2023.06.27.546055 (2023) https://doi.org/10.1101/2023.06.27.546055.

aMe26_R. There are 4 aMe12s on each side of the brain, and KCγ-d's contact all 8 individual neurons (7 with more than 5 synapses, 1 with fewer). KCγ-d's contact 5/6 aMe26 neurons in the brain. There is only one aMe20 in each hemisphere. Therefore, KCγ-d's contact most individual members of these cell types.

As a group, the aMe12 neurons are the top inputs to KCγ-d's; collectively, they contact almost half of all KCγ-d's and provide them with the most synaptic input out of any VPN type (Fig. 2F, G). aMe12 neurons receive direct input from photoreceptor cells[16], and thus their inputs to Kenyon cells are of the same circuit depth as olfactory projection neurons inputs, i.e., two synapses from the periphery. Most other paths to KCγ-d's appear to be three or more synapses from the periphery. Interestingly, aMe12 specifically receives excitatory input from the pale subtype of R7/R8 photoreceptors which expresses rhodopsins sensitive to short wavelength ultraviolet (R7) and blue (R8) light[16,50–52]. This

suggests that most KCγ-d's would be responsive to UV/blue light. A previous study found that in addition to blue light, KCγ-d's can also respond to green light[15]. It is not clear which cells in our dataset could be mediating Kenyon cell responses to other colors, or to what extent Kenyon cells may be preferentially tuned to specific wavelengths.

Generally, there is a clear absence of retinotopic columnar cell types among VPNs contacting the vACA. They are all large field neurons and many of them have dendrites restricted to the ventral part of the medulla. All medulla and lobula neurons are ipsilaterally connected. aMe12 and aMe26_R are the only direct VPNs connected both ipsi- and contralaterally. The top types of VPNs from each neuropil dominate synaptic input to KCγ-d's overall (Fig. 2E). As we will describe further below, these types dominate mushroom body inputs more prominently than do top olfactory types. We ranked VPN types by the sum of synapses they make onto KCγ-d's (Fig. 2F). Neurons from all

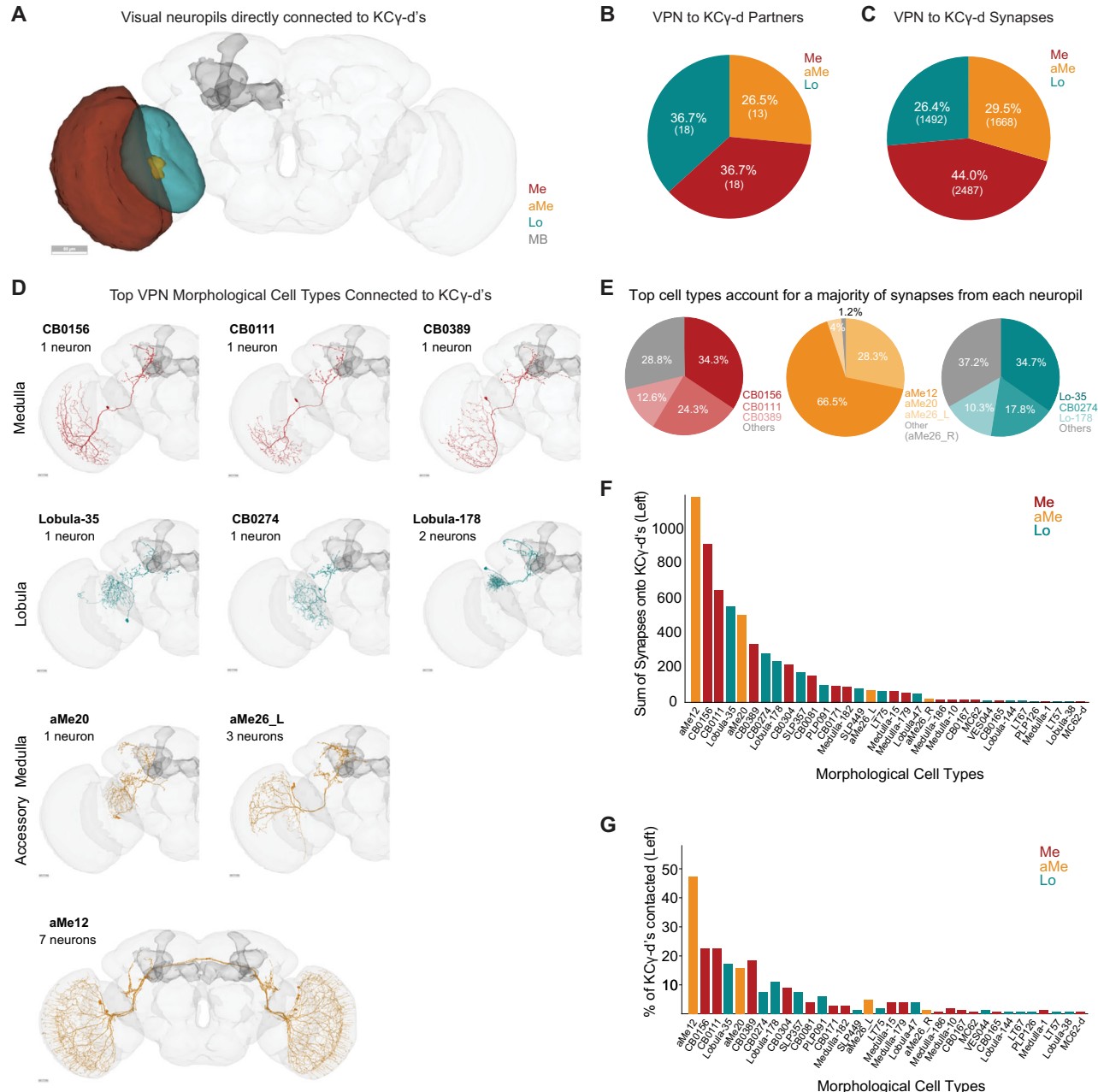

**Fig. 2 | Direct visual inputs to KCγ-d neurons come from diverse optic lobe neuropils. A** Schematic of optic lobe neuropils and mushroom body in the left hemisphere. The same neuropils are present in the right hemisphere but are not highlighted here. Me, Medulla; aMe, Accessory Medulla; Lo, Lobula; MB, Mushroom Body. Scale bar, 50μm. **B** Proportion of KCγ-d VPN partners that originate from one of the 3 optic lobe neuropils in (**A**). **C** Proportion of VPN synapses onto KCγ-d by optic lobe neuropil origin. **D** Morphologies of the top three VPN classes from each neuropil that provide the most synaptic input to KCγ-d's. Scale bars, 20 μm. **E** Proportion of synapses from each neuropil that come from the top 3 morphological types in (**D**). **F** Ranking of morphological VPN classes by total synaptic contribution to KCγ-d's. **G** Percentage of KCγ-d's contacted by each morphological

VPN class (out of 147 KCγ-d's in the left hemisphere). VPN classes are ordered on the x-axis as in (**F**) to show that classes making larger numbers of synapses also contact larger percentages of KCγ-d partners. Data in this figure examine connectivity to KCγ-d in the left hemisphere, using a ≥ 5 synapse threshold. Source data are provided as a Source Data file. Figure 2A, D neuron skeletons and neuropil volumes adapted with permission under CC BY-NC 4.0 license (https://creativecommons. org/licenses/by-nc/4.0/) from Dorkenwald et al. Neuronal wiring diagram of an adult brain. bioRxiv 2023.06.27.546656 (2023) doi:10.1101/2023.06.27.546656 and Schlegel et al. Whole-brain annotation and multi-connectome cell typing quantifies circuit stereotypy in *Drosophila*. bioRxiv 2023.06.27.546055 (2023) https://doi.org/ 10.1101/2023.06.27.546055.

three visual neuropils are among the top 5 input types, suggesting that signals from each neuropil are likely to have a strong influence on KCγ-d activity. The strongest connected neurons are also the ones that contact the most Kenyon cells (Fig. 2F, G).

As can be seen in the neuronal skeletons in Fig. 2D, every VPN innervating the mushroom body calyx also projects to other regions of the brain. Their processes in the central brain make a large number of

connections with other nearby regions, such as the posterior lateral protocerebrum (PLP) or superior lateral protocerebrum (SLP), which are brain areas implicated in sensorimotor decision-making and multisensory convergence[53–55]. Similar to VPNs, uniglomerular olfactory projection neurons that provide input to the mushroom body extend a second axonal branch to the lateral horn where they regulate innate behaviors[26,56].

## LVIN inputs to the ventral accessory calyx provide structured visual inputs

In addition to direct VPNs, KCγ-d's also receive visual inputs indirectly through LVINs (Fig. 1A, C). These are central brain interneurons with no arborizations in the optic lobe but that themselves receive inputs from VPNs[7]. We have identified 25 LVINs in the left hemisphere of the FlyWire reconstruction that make 5 or more synapses with KCγ-d's, corresponding to 21 cell types (Fig. 1E, Fig. 3A). The highest connected LVIN, PLP095 (PLP.78 in FlyWire), makes 438 synapses with a total of 24 KCγ-d's, while the median LVIN makes 11 synapses across 2 KCγ-d's. 19 of the LVINs are predicted to be cholinergic (i.e., excitatory) while 5 are predicted to be GABA/glutamatergic (inhibitory; Fig. 3C). These neurons also receive diverse inputs from other central brain neurons, including multiglomerular PNs, making these potential nodes of multisensory integration (Supplementary Fig. 3B).

Each LVIN receives inputs from multiple VPNs, which can have different neuropils of origin (Supplementary Fig. 3A-C), and the same VPN can make connections with multiple LVINs (Fig. 3F). We identified 226 indirect VPN inputs that make 5 or more synapses with KCγ-d-connected LVINs (Fig. 3A, F). A striking finding is that, among them, we find 36 out of the 49 direct VPNs described in Fig. 2 (Fig. 3B). In particular, all three direct aMe types connect with KCγ-d's both directly and indirectly through LVINs (Fig. 3F). Additional aMe neurons (aMe3, aMe5, aMe19a, and aMe25) are included in the indirect pathway. A small number of lobula plate neurons and ocellar neurons also provide inputs to LVINs. Despite not being represented in the direct VPN class, several types of small-field neurons whose dendrites tile the extent of the medulla/lobula, such as lobula columnar neurons (LCs) or medulla columnar neurons (MCs), provide inputs to LVINs contacting KCγ-d's (Fig. 3F, Supplementary Fig. 2A, F). When only inspecting neurons that make ≥5 synapses, we noticed that not all members of an LC or MC cell type are connected to an LVIN, giving rise to a seemingly random salt-and-pepper organization of receptive fields (Supplementary Fig. 2B, C). However, when removing the synapse threshold, 49 out of 52 LC15 neurons, for instance, are connected to KCγ-d-projecting LVINs, encompassing the extent of the lobula (Supplementary Fig. 2B, D). This is also true of LC24 where 53/62 neurons are connected (Supplementary Fig. 2C, D). In the case of MC62, although removing the threshold does include additional neurons, they are mostly restricted to the ventral half of the medulla, leaving out the MC62s that occupy the dorsal half (Supplementary Fig. 2E–G).

The connectivity matrix of VPN inputs onto vACA-projecting LVINs (Fig. 3F) highlights interesting properties. The connectivity appears highly structured; certain VPN cell types make connections with few and particular LVINs (Fig. 3D, F). For instance, each individual neuron in the lobula columnar LC15 subtype only connects to either or both PVLP.SCL.2 (PLP009 in hemibrain) and PVLP.SLP.13 (SLP306 in hemibrain), and each of these two LVINs primarily receive inputs from this cell type and not from other VPNs (Fig. 3D, F). A similar situation exists in the case of MC62s: 97% of individual neurons of this type connect strongly to the LVIN PLP251, and 7% also connect to either PLP231 or PLP120 (Fig. 3D, F). In contrast to the LVINs downstream of LC15 that mostly receive input from LC15, the LVIN PLP251 also receives inputs from 19 other VPN types. In order to test for structure in these connectivity patterns in a more systematic way, we performed principal components analysis on the VPN-LVIN connectivity matrix (Fig. 3E). We found that some components accounted for additional structure when compared to shuffled matrices preserving VPN connection probabilities and the number of inputs to each LVIN, demonstrating that these connections are inconsistent with a random model. The structured connectivity between VPNs and LVINs makes important predictions in terms of the degree of translational invariance that such a system might afford (see Discussion).

## Receptive fields of neurons providing inputs to KCγ-d's

Some of the most distinctive features of visual neurons are the extent and location of their processes in the optic lobe neuropil they innervate. These are related to their receptive field size and position in the field of view of the animal. We used a newly developed eye map that allows the prediction of the receptive field of optic lobe neurons given the extent and location of their processes to extract receptive field size and position, quantify these properties and relate them to their connectivity with KCγ-d's (Fig. 4A)[57]. We first applied this methodology to direct and indirect VPNs from each neuropil to describe their individual receptive field properties and then extended our analysis to predict the visual receptive fields of individual KCγ-d's (Fig. 4B, Supplementary Fig. 4).

We found that for direct VPNs, especially in the case of medulla VPNs but also to a certain extent for aMe and lobula VPNs, the neurons that have the largest receptive fields are the ones that are the most strongly connected with KCγ-d's (Fig. 4B, C, Supplementary Fig. 4). Although it seems to be a general trend across the brain that visual projection neurons with larger receptive fields form more output synapses (Fig. 4E), we found that the specific subsets of Me/aMe/Lo VPNs directly synapsing onto the KCγ-d population have on average larger receptive fields when compared to the full VPN population (Fig. 4F). This property holds to a lesser extent for indirect VPNs (Fig. 4C, right). We next computed the location of the centroid of the receptive field of each VPN and plotted them on the eye map, as viewed by the left eye of the animal, weighted by the number of connections made by each VPN onto Kenyon cells (for direct VPNs) or by each VPN onto LVINs (for indirect VPNs; Fig. 4D, left; weight is indicated by the size of the dot). We found that highly connected direct medulla VPNs preferentially represent the ventral part of the field of view of the fly, whereas highly connected direct lobula VPNs represent the dorsal part. Indirect lobula and medulla VPNs do not show this dorsoventral distinction. Indirect aMe VPNs, however, preferentially represent the dorsal half of the field of view. We also noticed a slight rostral shift of lobula VPN centroids as compared to medulla VPNs, though this may reflect a technical limitation of the prediction tool. These properties are also apparent when overlaying the receptive fields of direct and indirect VPNs, separated between the three neuropils and weighted by the number of connections made by each VPN onto Kenyon cells or LVINs (shading; Fig. 4B).

Next, we used these data to derive the putative receptive field of single KCγ-d's by linearly combining incident VPN and LVIN receptive fields, weighted by synaptic count. We show representative examples in Fig. 4G. Overall, each KCγ-d samples most of the field of view of the animal and receives information from different parts of the visual field from different neuropils. The small size of the indirect VPN receptive fields is clearly apparent in some of these examples. The indirect receptive field composite images are combinations of LVIN receptive fields. From this analysis, we conclude that each KCγ-d receives mixed visual information, made up of combinations of VPNs, with direct VPNs encompassing larger parts of the field of view and indirect VPNs smaller parts of the field.

## Individual KCγ-d's select inputs randomly from a precise set of visual neurons with heterogeneous weights

We next extended our analysis of the sets of inputs onto individual KCγ-d's to the whole population. We generated a matrix of connections onto each individual Kenyon cell in order to assess whether KCγ-d's receive labeled line, combinatorial, or randomized inputs (Fig. 5A). Individual KCγ-d's receive 1–7 visual inputs each, with a median of 3 (Fig. 5B). Five identified KCγ-d's did not receive any inputs from VPNs or LVINs, and instead received input from olfactory PNs (Fig. 5B). This median of 3 visual inputs per KCγ-d is slightly lower than the median 5-6 inputs received by olfactory Kenyon cells in general and less than the 4–8 inputs that their closest developmental sisters, olfactory KCγ-

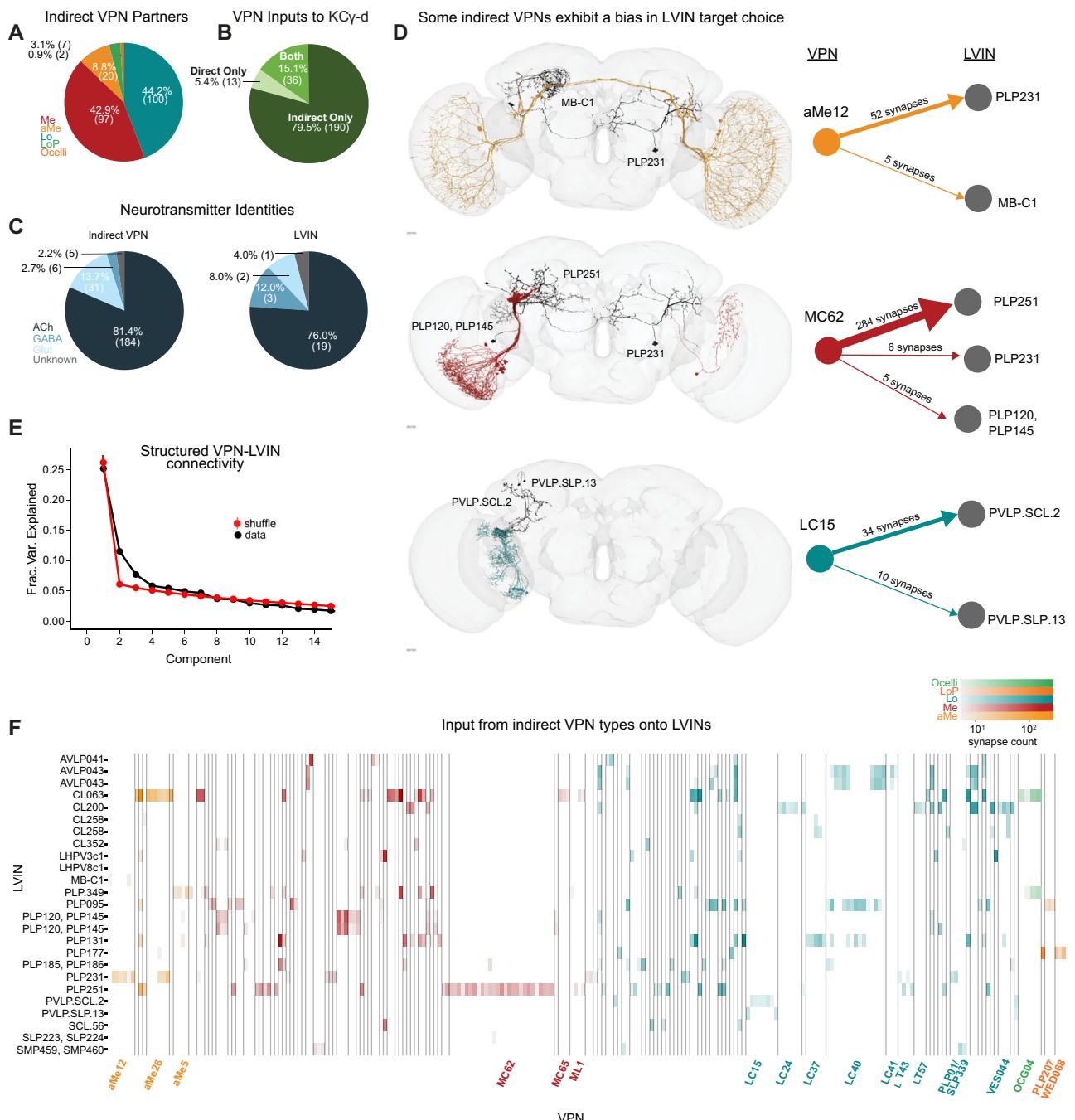

**Fig. 3 | LVINs relay structured visual inputs to KCγ-d. A** Proportion of indirect VPNs originating from a given neuropil. Me, Medulla; aMe, Accessory Medulla; Lo, Lobula; LoP, Lobula Plate. **B** Proportion of VPNs in our dataset providing direct, indirect (via LVINs), or both types of input to KCγ-d's. **C** Predicted neurotransmitter identities (Eckstein, Bates, et al., 2023) for indirect VPNs (left) and LVINs (right). ACh, acetylcholine; Glut, glutamate. **D** Indirect VPN examples that demonstrate a bias for certain LVINs. aMe12s (top) synapse with LVIN PLP231, with one aMe12 synapsing with MB-C1. Most MC62s (middle) converge onto the LVIN PLP251. One MC62 from the right hemisphere contacts PLP231, which synapses onto KCγ-d in the left hemisphere. LC15s synapse onto PVLP.SCL.2 and PVLP.SLP.13. Scale bars 15 μm. **E** Principal components analysis of indirect VPN to LVIN connectivity. Red circles and bars represent mean and 95% confidence intervals for the variance explained by the principal components of shuffled connectivity matrices where VPN connection probabilities and number of inputs to each LVIN are preserved

($N = 1000$ shuffles). Top PC components account for more structure than shuffled matrices. **F** VPN-LVIN connectivity matrix. Individual LVINs and LVIN classes receive structured inputs from specific VPN classes. Indirect VPN classes that correspond to previously named morphological types and contain three or more neurons are labeled. Data in this figure examine connectivity to KCγ-d in the left hemisphere, using a ≥ 5 synapse threshold. Source data are provided as a Source Data file. Figure 3D neuron skeletons and neuropil volumes adapted with permission under CC BY-NC 4.0 license (https://creativecommons.org/licenses/by-nc/4.0/) from Dorkenwald et al. Neuronal wiring diagram of an adult brain. bioRxiv 2023.06.27.546656 (2023) https://doi.org/10.1101/2023.06.27.546656 and Schlegel et al. Whole-brain annotation and multi-connectome cell typing quantify circuit stereotypy in *Drosophila*. bioRxiv 2023.06.27.546055 (2023) https://doi.org/10.1101/2023.06.27.546055.

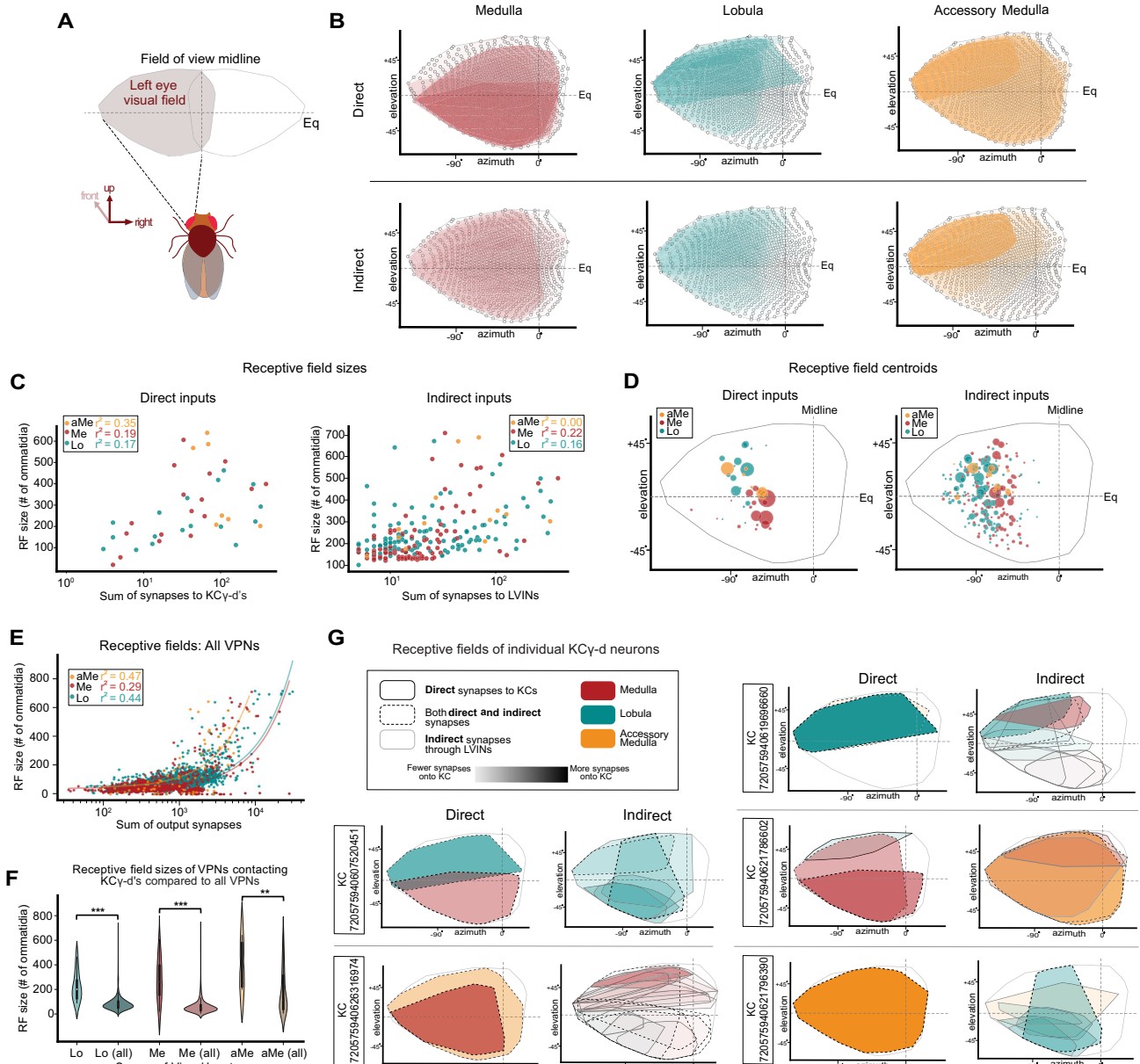

**Fig. 4 | Receptive fields (RFs) of direct and indirect VPNs. A** Region of visual space sampled by left eye ommatidia. Regions above (elevation > 0°) and below the equator (elevation <0°) represent the dorsal and ventral visual fields, respectively. **B** Top, estimated RFs of direct VPNs are overlaid. Darker shading, more synapses onto KCγ-d's. Bottom, estimated RFs of indirect VPNs are overlaid. **C** Left, RF sizes of direct VPNs are plotted against the synapse sum onto KCγ-d's. Right, RF sizes of indirect VPNs are plotted against the sum of synapses to LVINs. Pearson correlation coefficients for each neuropil are reported. **D** Left, centroids of direct VPN RFs. Circle size is proportional to total synapses made with KCγ-d's. Right, centroids of indirect VPN RFs. Circle size is proportional to total synapses onto LVINs. **E** Approximate RF sizes of all VPNs from the left optic lobe. RF sizes are plotted against the sum of all output synapses. Correlation coefficients for each input neuropil are reported. **F** RF sizes for VPNs contacting KCγ-d's in the left hemisphere

(Lo ($N = 18$; center=200.5, lower bound=128, upper bound=272.5, lower whisker=90, upper whisker=463), Me ($N = 18$; center=260, lower bound=157.5, upper bound=393.25, lower whisker=22, upper whisker=606), aMe ($N = 7$; center=251, lower bound=224.5, upper bound=577, lower whisker=202, upper whisker=639)) and all VPNs in the brain receiving left optic lobe input (Lo ($N = 2849$; center=73, lower bound=52, upper bound=100, lower whisker=7, upper whisker=713), Me ($N = 956$; center=45, lower bound=31, upper bound=71, lower whisker=7, upper whisker=707), aMe ($N = 65$; center=113, lower bound=41, upper bound=312, lower whisker=8, upper whisker=639)). All samples are biologically independent cells examined over one independent experiment. Two-sided Student's t-test: $p = 1.734e-16$ (Lo), $p = 1.90e-23$ (Me), $p = 0.007$ (aMe). **G** Effective RFs of five representative KCγ-d's. Left, direct inputs, Right, indirect inputs. Data examine the RF properties of VPNs from the left optic lobe. Source data are provided as a Source Data file.

main, receive in this brain[8,28,47]. We next analyzed the origin of inputs onto each cell (medulla, lobule, accessory medulla or LVIN). Strikingly, all distributions of inputs were possible, from KCγ-d's receiving pure input from one neuropil to every mixture in between (Fig. 5C).

Next, we asked whether specific inputs converged onto specific KCγ-d's by clustering KCγ-d's according to the inputs they received (Fig. 5D, Supplementary Fig. 5C). We did not see evidence of sorting

or convergence in this data: KCγ-d's did not fall into obvious groups and knowing one input to a particular KCγ-d was not informative about other inputs. Hierarchical clustering performed on the matrix of connections onto KCγ-d's did not reveal any apparent structure, similar to the same procedure applied to KCγ-m's, which receive distributed, combinatorial input from uniglomerular olfactory projection neurons (Supplementary Fig. 5B, F). This contrasts with

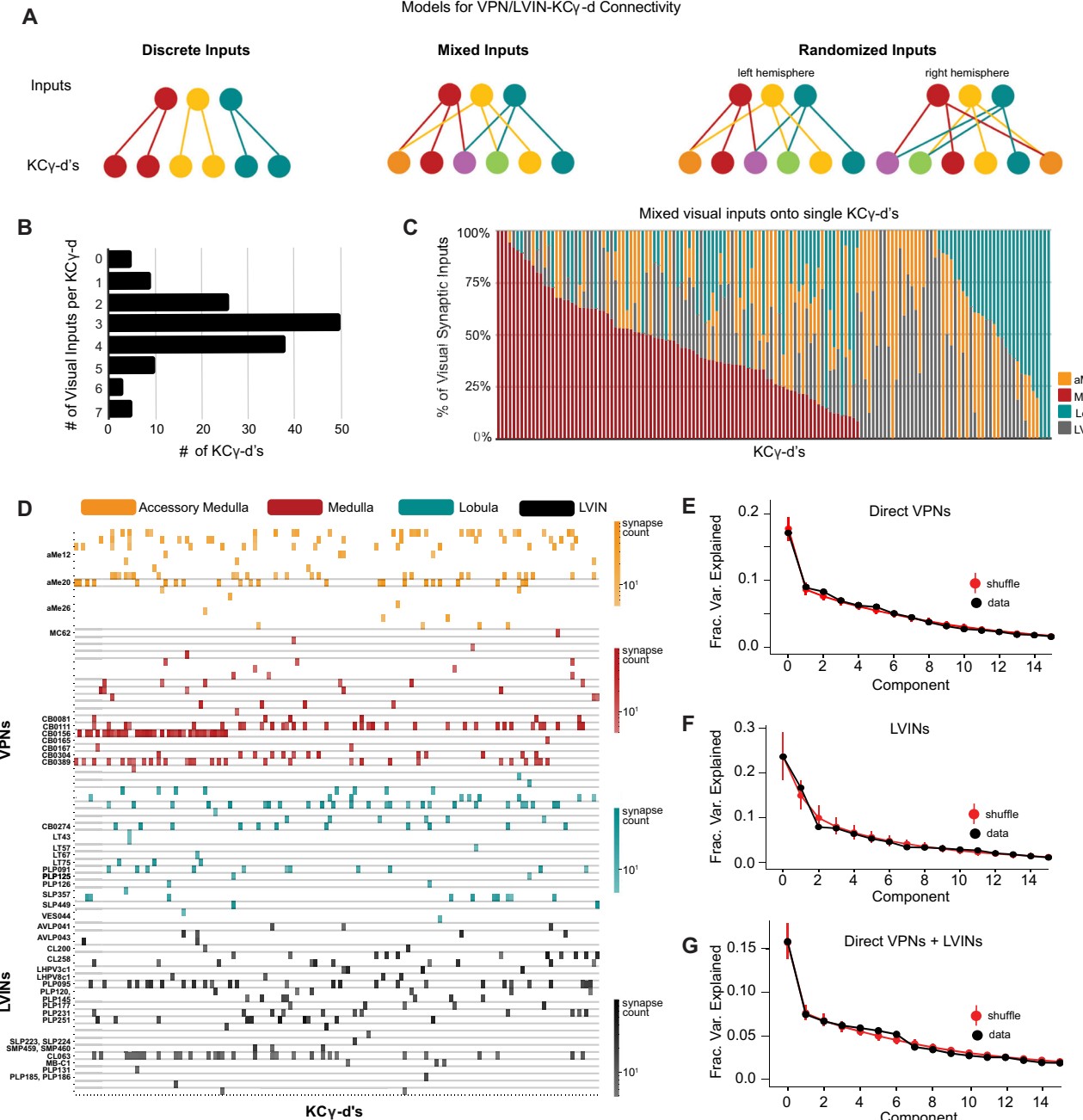

**Fig. 5 | VPNs and LVINs make distributed, combinatorial inputs onto KCγ-d's.**
**A** Models illustrating potential connectivity patterns between VPNs/LVINs and KCγ-d's. Left, discrete visual channels; Middle, mixtures of visual inputs onto KCγ-d's; Right, random mixtures of visual inputs onto KCγ-d's, examined across hemispheres. The statistical tests used (see "Methods") to ascertain the presence of structured input onto KCγ-d's are sensitive enough to detect multiple types of structure beyond what is illustrated here. **B** Individual KCγ-d's (n = 147) receive 1–7 unique inputs from VPNs/LVINs, median = 3 inputs. The 5 KCγ-d's that receive 0 VPN inputs receive inputs from olfactory projection neurons. **C** KCγ-d's receive mixtures of visual inputs. Columns, synaptic input compositions of individual KCγ-d's (n = 142 KCγ-d's, i.e., the ones which receive visual input). X-axis is ordered by KCγ-d's receiving the highest proportion of Me input left to right and then by KCγ-d's receiving the highest proportion of Lo input right to left. **D**. Connectivity from direct VPNs and LVINs onto KCγ-d's. VPNs are sorted by input neuropil and morphological type. LVINs are sorted by neurotransmitter and morphological type.

KCγ-d's were sorted using spectral clustering (see "Methods"). **E** Principal components analysis of direct VPN to KCγ-d connectivity. Red circles and bars represent mean and 95% confidence intervals for variance explained by the principal components of shuffled connectivity matrices (N = 1000 shuffles). Top PC components do not account for higher proportions of variance than shuffled matrices preserving VPN connection probability and number of inputs to each KC. **F** Principal components analysis of LVIN to KCγ-d connectivity. Red circles and bars represent mean and 95% confidence intervals for variance explained by the principal components of shuffled connectivity matrices (N = 1000 shuffles). **G** Principal components analysis of direct VPNs and LVINs to KCγ-d connectivity. Red circles and bars represent mean and 95% confidence intervals for variance explained by the principal components of shuffled connectivity matrices (N = 1000 shuffles). Data in this figure examine connectivity to KCγ-d in the left hemisphere, using a ≥ 5 synapse threshold. Source data are provided as a Source Data file.

hierarchical clustering of inputs to individual olfactory projection neurons, which each receive convergent and deterministic input from dedicated olfactory sensory neurons in the antennal lobe (Supplementary Fig. 5A, E).

A variety of methods have demonstrated the inputs to olfactory Kenyon cells in the adult are largely consistent with a random sampling of locally available olfactory projection neuron boutons[7,28,58], though, in the ventral olfactory calyx, KC-αβ's receive biased input from specific PN types[59,60]. To further test for structure in the visual inputs to KCγ-d's, we examined the spectrum of the principal component eigenvalues of the Kenyon cell input covariance matrix, as has been done previously (Fig. 5E–G)[7,13,28]. Comparing this spectrum to randomly shuffled controls provides a sensitive measure of whether particular patterns of input are overrepresented across the Kenyon cell population. We found no difference between the spectrum obtained from the EM reconstruction and shuffled spectra, consistent with random sampling of visual inputs. By comparison, the same methods identified clear structure in the highly stereotyped olfactory receptor neuron to olfactory projection neuron connectivity (Supplementary Fig. 5E) and a smaller amount of structure in olfactory projection neuron to KCαβ/KCα'β' connectivity (Supplementary Fig. 5F, G) consistent with prior studies. When inspecting loadings of the top PC components for VPN/LVIN input connectivity to KCγ-d's, we found that many inputs with the largest loading magnitudes also had the largest connection probabilities (Supplementary Fig. 7A), indicating that the residual structure observed in this data can likely be explained by biased input connectivity. By comparison, we found that the top PC components for VPN to LVIN connectivity (Fig. 3F) distinguish groups of input VPNs contacting specific target LVINs (Supplementary Fig. 7B), supporting the existence of input structure in this dataset.

To further support the claim that KCγ-d's receive random input, we used the conditional input analysis developed by Zheng et al.[59] (see "Methods"). Specifically, we counted the number of connections to the KC cell population from one input A given input from another input B for all possible pairs of input. For each pair of inputs, we then calculated a z-score from a null distribution computed from randomly shuffled connectivity matrices to generate a "conditional input matrix" and determine if observed connectivity from a specific input was significantly more or less dependent on connectivity from a second input than would be expected in a random model of connectivity. To extract groupings of inputs based on these conditional input scores, we then applied unsupervised K-means clustering to the conditional input matrix[61]. We found that, in alignment with our own PCA-based randomness analyses, we could not identify any significant structure in the clustered p-value matrix from a conditional input analysis applied to direct VPN and LVIN connectivity to KCγ-d's (Supplementary Fig. 6B), whereas we could extract a small amount of structured connectivity relating to a subset of ventrally-projecting PNs (as reported by Zheng et al.[59]) from a conditional input analysis applied to olfactory PN connectivity to KCs (Supplementary Fig. 6A).

Although these results support random sampling of visual input, we find that the inputs to visual KCγ-d's are more skewed toward the top few input types than are olfactory inputs to Kenyon cells (Supplementary Fig. 5D). To quantify this observation, we computed the predicted dimension of the visual input currents received by KCγ-d's, as quantified by the participation ratio[34,62], and compared it to that of the olfactory inputs to KCγ-m's. The participation ratio was lower in the case of visual connectivity compared to olfactory connectivity (visual: PR = 9.1, olfactory: PR = 15.4) despite the greater number of visual input channels, suggesting that visual inputs are dominated by a small number of input dimensions.

Nearly all uni-glomerular olfactory projection neurons target the olfactory calyx with a probability of forming connections onto Kenyon cells that depend on glomerular and Kenyon cell types consistently across individuals within a species[7,8,30,31,60,63]. VPNs and LVINs are far more developmentally diverse than are olfactory projection neurons, and the neurons we identify projecting to the left vACA in FlyWire are only a tiny fraction of all VPNs (49 out of 3933 total left VPNs, ~1.2%). To ask if the same VPN and LVIN sets target the vACA between hemispheres, i.e., if this aspect of development is predictable, we generated a map of visual inputs to KCγ-d in the right hemisphere (Fig. 6B). The top inputs to the left and right hemispheres were the same, and neurons identified as KCγ-d inputs in only one hemisphere were of low rank in the hemisphere where they did connect (Fig. 6A, B). Nevertheless, for 30/154 cell types that provided at least five synapses to the vACA in one hemisphere, we found that the paired cell type in the other hemisphere provided no calyx input, even with the synapse threshold removed. These cells may be a source of inter-hemisphere or inter-individual variation in learnable visual signals.

To examine whether there is stereotypy in the specific combinations of these inputs that KCγ-d's in the left and right hemispheres sample, we asked whether there is an overrepresentation of left-right KCγ-d pairs that receive input from homologous visual inputs, as was previously done for olfactory inputs to larval KCγ-d's[13]. Of 147 KCγ-d cells in the left hemisphere, only 15 had a cell in the right hemisphere that received an identical set of visual inputs, no more than expected from shuffled connectivity matrices in which Kenyon cells in each hemisphere sample their inputs independently (Fig. 6C). In comparison, pairs of olfactory projection neurons receiving identical input from homologous olfactory receptor neuron input types in the left and right hemisphere were significantly overrepresented, confirming high stereotypy between hemispheres for this stage of olfactory processing (Fig. 6D). The principal component eigenvalues for the combined left-right input connectivity matrix also were not significantly different from those of shuffled models (Fig. 6E).

Finally, to examine inter-individual variability, we repeated these analyses for homologous neurons identified in the hemibrain dataset. We used NBLAST[64] to query the morphology of each VPN and LVIN in our dataset against all neurons in the hemibrain volume (see "Methods" and results in Source Data files). Briefly, NBLAST is an algorithm that measures the similarity of two neuron structures, assigning them a similarity score between 0 and 1 (0=different neurons, 1=identical). We found confident matches in the hemibrain for 66% of VPN/LVINs in our dataset. These represent pairs of neurons with NBLAST scores greater than 0.5 (Fig. 7A). We found that the number of synapses formed by VPNs and LVINs onto KCγ-d's are highly correlated across the two datasets (Fig. 7B, C). As expected, principal components analysis found no evidence for stereotypy in these connections (Fig. 7D). In conclusion, our analyses suggest that the connectivity between LVINs/VPNs and KCγ-d's is consistent with random sampling, but strongly biased toward specific input cell types, which account for large proportions of visual information flow to these Kenyon cells.

## Discussion
### Striking heterogeneity in the visual inputs to the mushroom body

The insect mushroom body has long been studied for its role in chemosensory perception and associative odor learning, and as such, the underlying circuits are well-described anatomically and functionally. Here we add to the rich knowledge of mushroom body olfactory coding by providing a complete anatomic description of an alternate sensory pathway to the adult mushroom bodies. Using the FlyWire electron microscopy reconstruction of the entire female adult fly brain, we have described all direct and indirect (via LVINs) visual inputs from the optic lobes to the ventral accessory calyx. By comparing the architectures of the olfactory and visual mushroom body circuits, we highlight potential areas of functional similarity and difference.

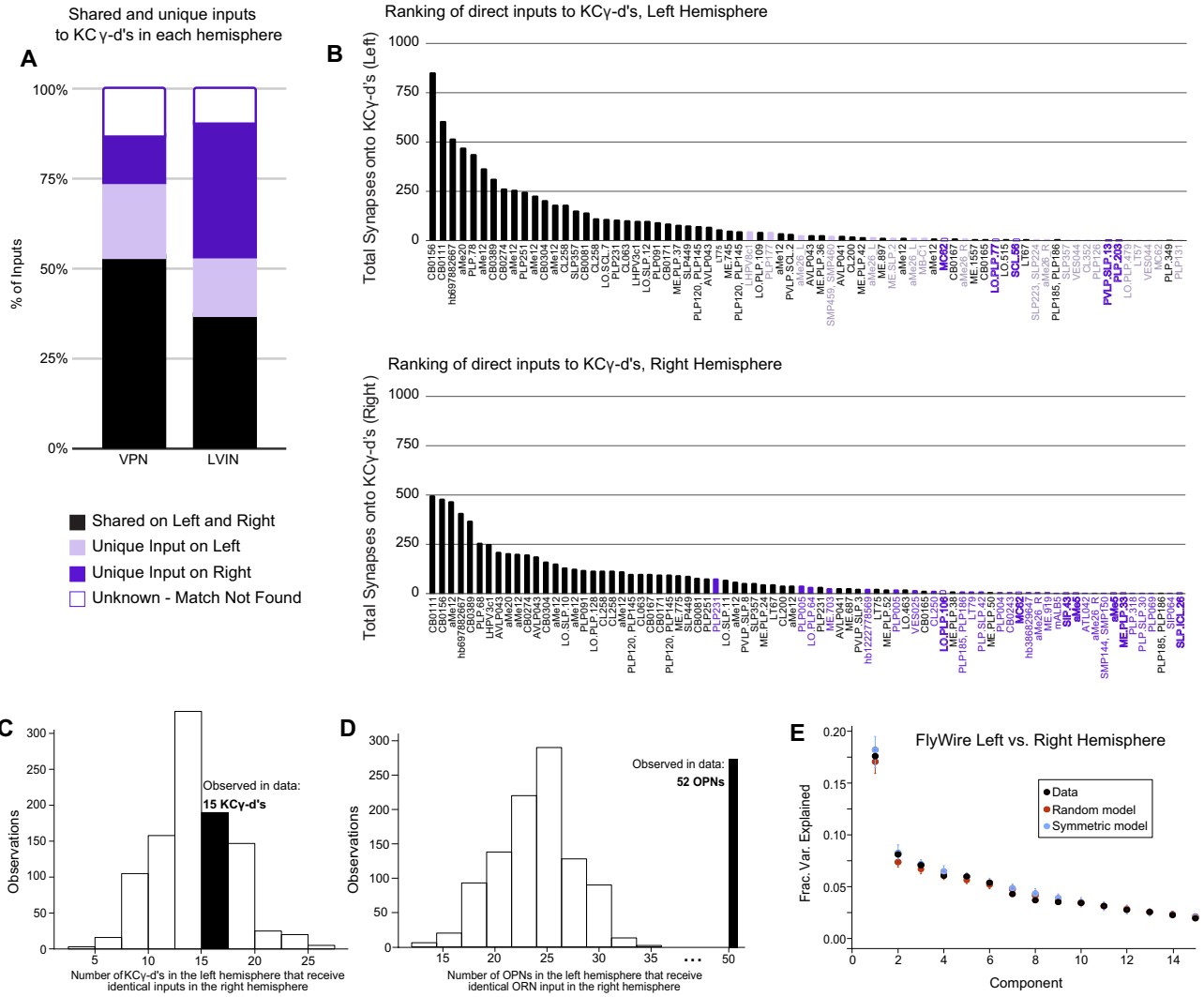

**Fig. 6 | KCγ-d's receive a random set of inputs from VPNs and LVINs. A** subset of VPN and LVIN inputs in the left hemisphere (black) have a mirror copy in the opposing hemisphere that synapses with KCγ-d ($n$ = 33 VPNs, 16 LVINs). Some VPNs and LVINs are unique inputs to either the left (light purple; $n$ = 13 VPNs, 7 LVINs) or right hemisphere (dark purple; $n$ = 8 VPNs, 16 LVINs). For a few neurons, no mirror copy could be easily identified (white box, purple outline; $n$ = 8 VPNs, 4 LVINs). **B** Ranking of inputs to KCγ-d in left (top) and right (bottom) hemispheres. Neuron names across the x-axis are colored as in (**A**). **C** Histogram of KCγ-d's in the left hemisphere that receive identical VPN/LVIN input patterns in the right hemisphere for 1000 randomly shuffled VPN/LVIN to KCγ-d matrices. In the FlyWire dataset, 15 KCγ-d's in the left hemisphere receive identical VPN/LVIN input patterns in the right hemisphere (black bar). **D** Histogram of OPNs in the left hemisphere with a counterpart in the right hemisphere receiving identical ORN input patterns (connectivity

was averaged over the same ORN type) for 1000 randomly shuffled ORN to OPN connectivity matrices. In the FlyWire dataset, 52 OPNs in the left hemisphere receive identical ORN input patterns in the right hemisphere (black bar). **E** Principal components analysis on the combined left and right hemisphere VPN/LVIN to KCγ-d connectivity matrix (black). Red circles and bars represent mean and 95% confidence intervals for the variance explained by principal components of a random model in which the connectivity matrix was shuffled, preserving VPN/LVIN connection probabilities and number of inputs to each KC ($N$ = 1000 shuffles). Blue circles and bars represent mean and 95% confidence intervals for the variance explained by principal components of a bilaterally symmetric model in which the connectivity matrix was random but identical for each hemisphere ($N$ = 1000 shuffles). Source data are provided as a Source Data file.

Indeed, there are similarities between the olfactory and visual mushroom body circuits, including the distributed and random sampling of sensory inputs by Kenyon cells. Dissimilarities are most obvious when examining the heightened level of cellular diversity among the visual inputs and hint at possible functional differences in the way sensory modalities are put to use in the mushroom body. The olfactory projection neurons connected to the mushroom body receive inputs in the antennal lobe and are thus all positioned one synapse from the periphery. Olfactory projection neurons are born from dedicated stem cells and follow conserved tracts from the antennal lobe to the mushroom body calyx[65–68]. In contrast, the visual PNs that we have described here originate from multiple optic lobe neuropils. While the aMe12 neurons receive direct input from

photoreceptors[16], medulla and lobula VPNs as well as LVINs are all at various depths of visual processing. Furthermore, single visual Kenyon cells can receive inputs from diverse mixtures of VPNs and LVINs; as a result, single visual Kenyon cells integrate information from various stages of visual processing. This anatomic arrangement differs from a model proposed in bees in which optic lobe sources are sorted into different calyx regions[69].

VPNs make up a majority of the direct inputs to KCγ-d's; however, about a quarter of the synapses to KCγ-d's (and the majority of synapses to KCαβ-p's) are from LVINs that provide indirect visual input to the mushroom body. Many of the LVINs were first described by Li et al.[7], and here we identified the full repertoire of VPNs that are routed to the mushroom body via LVINs. LVINs integrate inputs from a

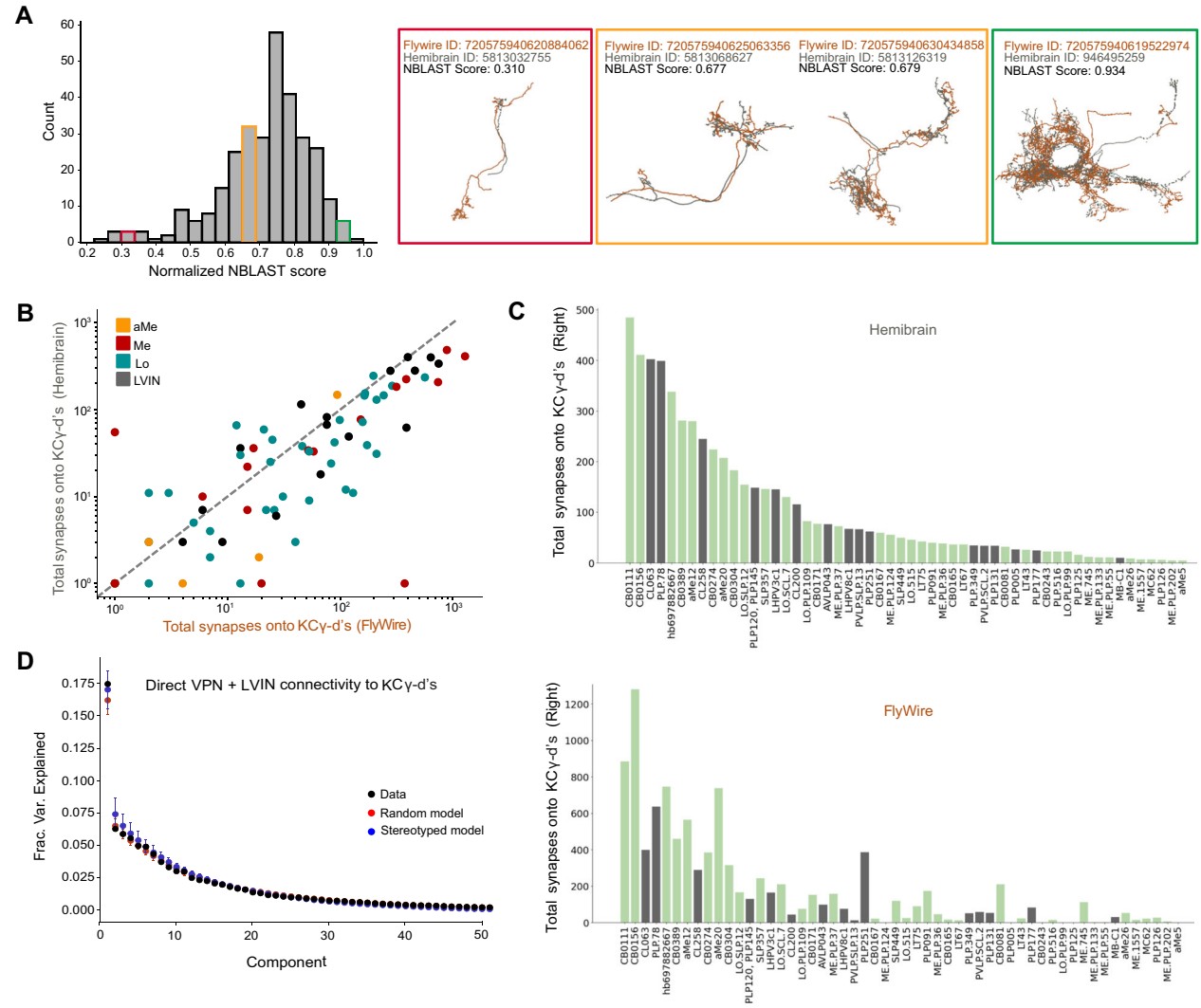

**Fig. 7 | Prominent visual inputs to KCγ-d are shared across brains.** **A** Left, distribution of NBLAST scores (normalized) when matching each FlyWire VPN or LVIN in one hemisphere to the full hemibrain dataset (see "Methods"). Right, representative examples of matched neurons with scores in three different NBLAST score ranges. Orange, FlyWire neuron skeleton. Gray, Hemibrain neuron skeleton. **B** For each VPN and LVIN in the FlyWire dataset, the sum of synapses onto KCγ-d's in FlyWire (*x-axis*) is plotted against the sum of synapses its match makes onto hemibrain (*y-axis*). Neurons that have higher connectivity onto KCγ-d's in hemibrain also tend to have higher connectivity onto KCγ-d's in FlyWire. **C** Top matched neurons in hemibrain are ranked by the sum of synapses made onto the KCγ-d population. Bottom, the sum of synapses onto the KCγ-d population for the FlyWire neuron matches the hemibrain neurons in the top panel; the order is consistent with the top panel to highlight that neurons that have larger weights are largely

common between the two brains. In total, FlyWire neurons tend to make more synapses onto KCγ-d's, and this is likely because there are more KCγ-d's in the FlyWire brain than in the hemibrain (148 vs. 99 KCγ-d's, right hemisphere)[47]. **D** Principal components analysis on VPN/LVIN connectivity to KCγ-d's (black) aggregated over both connectomes. Red circles and bars represent mean and 95% confidence intervals for the variance explained by principal components of shuffled connectivity matrices preserving VPN/LVIN connection probabilities and a number of inputs to each KC ($N = 1000$ shuffles). Blue circles and bars represent mean and 95% confidence intervals for the variance explained by principal components of bilaterally symmetric connectivity matrices in which VPN/LVIN-KC connectivity is random but identical in both connectomes ($N = 1000$ bilaterally symmetric matrices). Source data are provided as a Source Data file.

large and diverse array of VPNs, especially when compared to VPNs that serve as direct inputs to Kenyon cells. For example, in the left hemisphere alone, there are 49 VPNs that connect directly to KCγ-d's and 190 additional VPNs whose signals are routed to KCγ-d's via LVINs. Interestingly, 36/49 direct VPNs synapse with an LVIN, taking an indirect route to the mushroom body as well. The dual visual input from direct and indirect sources raises an interesting question about the functional role of each pathway type, and why most direct VPNs are also processed in parallel via the indirect pathway. Nevertheless, it is clear from an anatomical perspective that the LVINs are integrators of visual input, and in the future it will be interesting to further explore their functional role in visual processing.

### What are Kenyon cells "seeing" and what are they using it for?
The similarity between the top VPN inputs to left and right hemispheres in the FlyWire dataset, as well as their similarity with VPNs in hemibrain, suggests that specific visual information is selected to be conveyed to the mushroom body. Unfortunately, the great majority of the VPNs in question have not been studied, making it difficult to assign functions. Their anatomical characteristics, however, provide some indications as to stimulus preference. Many of the direct VPNs that we describe here sample broadly across the visual field, supporting the notion that this circuit is adapted for processing general features of a visual scene (e.g., luminance or color) rather than for encoding precise spatiotemporal relationships[70]. An interesting

example is that of large-field aMe neurons, which make up 29.5% of VPN synapses onto KCγ-d's. While the role of aMe neurons has not been functionally determined, given their position as inputs to circadian neurons, it is likely that they could provide context to the mushroom body with information related to the time of day[71]. Despite the lack of overt spatial organization of the visual inputs in the mushroom body, there is precedent in other model systems for spatial encoding of stimuli in non-retinotopic circuits, which underscores the need for functional validation of these anatomic data[72]. Experiments aimed at probing the stimulus selectivity of VPNs and Kenyon cells will be necessary to clarify the stimulus space to which the mushroom body is tuned.

In contrast to the direct inputs, LVIN inputs to KCγ-d's process information from six kinds of small-field lobular columnar (LC) neurons, amongst others. LCs are the main columnar output of the optic lobe and are thought to each retinotopically encode different features of the visual scene[73–76]. We have found that most members of the LC15 type, previously shown to respond specifically to long thin moving vertical bars[73,74], make connections with 1-2 LVINs. This suggests that a subset of KCγ-d's, receiving inputs from these particular LVINs, will respond to this particular stimulus, regardless of its location in the field of view of the animal. This convergent arrangement is ideal for providing translational invariance to stimuli in the mushroom body and therefore may contribute to visual learning about specific kinds of objects regardless of their spatial position.

Given that KCγ-d's and KCαβ-p's represent only 8% of all Kenyon cells, what functional role might they play in the mushroom body, which is otherwise dominated by olfactory input? Visual Kenyon cells could be neural substrates for pure visual learning; they could be overshadowed[77] by olfactory Kenyon cells and serve little purpose; and/or they could provide context for olfactory learning. There is substantial evidence for the latter. For instance, when a visual stimulus is paired with an odor, flies memorize these multisensory cues at higher rates compared to when the visual or olfactory stimuli are presented alone, a phenomenon that relies on KCγ-d activity[41,42]. This role in multisensory learning is further supported by the inner circuitry of the mushroom body; in contrast to their separate inputs in the calyx, the axons of visual and olfactory Kenyon cells receive reinforcement from many common dopaminergic neurons (DANs) and synapse onto many common mushroom body output neurons (MBONs)[7,78]. Additionally, a recent study showed that the serotonergic neuron DPM forms an excitatory bridge between visual and olfactory Kenyon cells, broadening the memory engram after a multisensory experience to improve memory performance[42].

A purely visual role for Kenyon cells has not been ruled out. Whereas most MBONs receive a large majority of olfactory inputs, reflecting the predominance of olfactory KCs, MBON19 and so-called atypical MBON27 and MBON33 receive a majority of their inputs from visual KCs, providing a potential substrate for a dedicated visual association pathway[7]. However, there have been mixed results from studies that have sought to test the fly's ability to complete a mushroom body dependent visual associative learning task. In reinforcement paradigms where two color stimuli are presented, and one is paired with either a reward or punishment, flies display a weak preference for the conditioned color in subsequent test trials[15,79]. It is not clear whether the colors tested in these paradigms activate separate populations of VPNs and, consequently, different subsets of visual KCs. Few visual stimuli have been tested in pure visual learning, and since our analysis shows that very specific visual neural types make their way to the mushroom body calyx, it is possible that extending the type of visual stimuli tested in associative learning paradigms will reveal significant mushroom body dependent visual learning.

## Kenyon cells sample randomly from precisely allocated presynaptic inputs

The ~2000 Kenyon cells in the main olfactory calyx connect randomly to the ~50 types of olfactory projection neuron inputs[7,28,58]. The sparse and combinatorial nature of these connections expands the possible number of odors that can be encoded by the mushroom body[34]. As very different parameters compose a visual object vs. a smell (e.g., color, brightness, edges, line orientations, location, etc. versus molecular mixtures and concentrations), one may intuit that the encoding of a visual percept in the mushroom body would inherently require a different circuit configuration. Instead, we find from examining the visual mushroom body connectivity in multiple hemispheres that, like olfactory Kenyon cells, visual Kenyon cells randomly sample from incoming VPNs and LVINs. This is predictive of a scenario in which the combinatorial coding of visual signals is used to generate visual percepts, yet if and how this type of coding scheme could generate coherent representations of visual objects is still unclear.

While the manner in which Kenyon cells connect to VPNs is random, the subset of VPNs included in this circuit seems relatively deterministic. In FlyWire, there are 7,851 total neurons annotated as VPNs. We find that just 469 of these 7,851 are direct or indirect visual inputs to KCγ-d's in left and right mushroom bodies. Among VPNs and LVINs connected directly to Kenyon cells, the top partners provide an outsized proportion of synaptic input, and the sharp differences in weights of these versus other inputs are consistent across hemispheres and brains. Much of this dominant input originates from accessory medulla neurons. The propensity for particular visual neurons to target the vACA, while others do not, suggests there are specific developmental instructions allowing these neurons to come into contact with KCγ-d's; once they do, the Kenyon cells sample randomly among them, seemingly agnostic to further features of LVIN or VPN identity. The adult KCγ-d set that we analyze here are the very same neurons that receive randomized olfactory input in the early larva, as these neurons prune their dendrites during metamorphosis before forming new, visual connections that will be used in the adult[13]. Though the random selection of inputs is a common feature of these cells across developmental time, their chosen partners switch from olfactory PNs in the larval circuit to this precise and diverse set of visual inputs in adulthood. What directs certain VPNs to synapse in the mushroom body and how Kenyon cells randomly select among these available partners will be interesting areas of future research.

## Evolution of visual inputs to the mushroom body and cognition

Across different insects, the size of the Kenyon cell repertoire and the balance of sensory modalities providing their input are highly divergent. In particular, species of *Hymenoptera*, *Lepidoptera*, and *Blattodea* have evolved ~100-fold expansions of the mushroom body, and these expansions are correlated with increased visual input and striking visual cognitive abilities[9]. The increase in visual Kenyon cell number and microglomerular structures in these species[9,80–82] correlate with with visual cognitive abilities that surpass what has been observed in flies: Many hymenopterans and lepidopterans use visual navigation to forage and particular species have also been shown to count, to recognize the faces of individual conspecifics, to be able to learn based on observation and inference, and to recognize visual cue configuration and abstraction[80,83–101]. Recent work in the bee mushroom body also highlights how changes in the expression of immediate early genes and a gene encoding a dopamine receptor are correlated to visual learning performances[102–104]. However, the lack of genetic control in insects outside of *Drosophila* means a functional requirement for the mushroom body in the sophisticated types of visual cognition described above has been addressed only rarely[15,44–46].

We show here that in flies, the mushroom body receives very specific channels of visual information: In the 80% of KCγ-d-inputs provided by VPNs, neurons representing full-field signals are prominent, while the 20% of inputs from LVINs include neurons representing small objects or specific kinds of motion. Among the insects with impressive visually-mediated behaviors, there is a paucity of information about the exact neuron types and connectivity patterns that link the optic lobes to the mushroom bodies, though elaboration of visual Kenyon cell number in bees is proposed to occur together with re-routing of small-field visual signals to the mushroom body[105]. However, the overrepresentation of large-field inputs versus small-field inputs to the visual mushroom body that we observe in flies could be conserved across insects, and simply be mapped onto 500 times more visual Kenyon cells in Hymenopterans. In the future, comparative connectomic studies of insects with variations in visually mediated behaviors could begin to reveal how the evolution of circuits correlates with the emergence of sophisticated visual cognition.

## Methods

### Identification and characterization of direct visual inputs to visual Kenyon Cells in FlyWire

Data and accompanying annotations from the FlyWire v630 snapshot were used unless stated otherwise[8,47–49,106–108]. Our analyses used a ≥ 5 synapse threshold when determining a connection between neurons unless stated otherwise, in keeping consistent with connections indicated in Codex (https://doi.org/10.13140/RG.2.2.35928.67844). KCγ-d and KCαβ-p neurons were found in the FlyWire database by searching in Codex for "hemibrain_type = KCg-d" or "KCab-p." To identify the direct visual inputs to these neurons, we used the natverse R package (R version 4.3.1 (2023-06-16), RStudio Version 2023.06.1 + 524) and accompanying fafbseg library[109]. Using the function flywire_partner_summary2() and specifying version = "630" we queried all inputs to visual KCs in the left and right hemispheres separately. We filtered the resulting inputs by neurons belonging to the "visual_projection" Super Class. Prior to filtering, we noticed that not all the upstream neurons returned by flywire_partner_summary2() were given a Super Class designation; we, therefore, searched for these unclassified neurons individually in Codex to find their Super Class and included them in our dataset if they were annotated as visual projection neurons.

Neuropil origins of the visual projection neurons (VPNs) were largely determined based on noting the prefix of the neuron's cell name in Codex as well as visually inspecting where in the optic lobe most dendrites for a neuron were located. aMe neurons were binned as such based on their given cell type classification in Codex.

Putative neurotransmitter (NT) identities for each neuron in our study were generated by Eckstein et al.[107] and listed in Codex. In the referenced study, an artificial neural network was trained to classify synapses in the fly brain as belonging to one of six transmitters (acetylcholine, GABA, glutamate, dopamine, serotonin, octopamine) based on differences in ultrastructural phenotypes. They report training the neural network with a ground truth data set of neurons with known neurotransmitter identity from both FAFB[8] and Hemibrain volumes[25] containing presynaptic sites that were manually annotated (FAFB) or automatically detected (Hemibrain). Using this method, they were able to classify the transmitter for a given presynapse with 88% accuracy in the FAFB volume and 79% accuracy in the Hemibrain volume. We report the predicted NT identity as was listed in Codex. Neurons predicted to be cholinergic generally had the highest reported confidence scores (>0.8 out of 1).

Neuron names are generally reported as the Hemibrain Type when listed in Codex. When Hemibrain Type information was not available, we either reported the name as the Cell Type or the temporary name (e.g., SLP.208). A full list of neurons included in this study can be found in the Source Data files.

During its acquisition, the EM volume in FlyWire was flipped[47]. We present the left and right FlyWire hemispheres as their corrected versions, as depicted in the Codex data explorer.

### Identification of LVINs and indirect visual inputs to Kenyon Cells in FlyWire

Local Visual Interneurons (LVINs) were first described by Li et al.[7]. They were defined as neurons that do not have processes in the optic lobes, receive input from VPNs, and synapse with Kenyon Cells. NBLAST[64] was initially used to map LVINs connected to KCγ-d's from the hemibrain to FlyWire. In FlyWire, all of these neurons belong to the "central" Super Class. We therefore searched for additional LVIN inputs to KCαβ-p's or KCγ-d's using the following filters: neurons belonging to the "central" Super Class (excluding Kenyon Cells, MBONs, DANs, APL, DPM, ALPNs), having ≥5 synapses with a single KCγ-d/αβ-p, and receiving input from visual projection neurons.

The indirect visual inputs to visual KCs were identified by querying the upstream inputs to each LVIN in Codex, filtered by "super_class = visual_projection". These cells were also filtered by having ≥5 synapses with a single LVIN unless stated otherwise.

When examining the resulting groups of indirect VPNs, we noticed that only a subset of some larger groups of neurons were synapsing with the LVINs. In the case of small-field Lobula Columnar neurons (LCs) which sample from discrete regions of visual space, this initially made it seem like an uneven and random assortment of visual space was being reported to the mushroom body. We checked if the ≥5 synapse threshold was masking the remaining inputs; we eliminated the synapse threshold while running flywire_partner_summary2(), querying inputs to LVINs without thresholding synapses. This revealed the remaining neurons that make 4 or fewer synapses onto LVINs (Supplementary Fig. 2).

### Identification of KCγ-d synapse types

Olfactory projection neurons have axon collaterals that form synaptic bouton structures in the main mushroom body calyx. These boutons are enwrapped by Kenyon Cell's dendritic claws. We examined the points of contact between VPN/LVIN axons and KCγ-d dendrites in the Codex 3D viewer and observed bouton-like terminals for both VPNs and LVINs that were enwrapped by KCγ-d dendrites. We designated these types of contacts as bouton-claw synapses. We also observed VPN/LVIN axons that were contacted along their length by a KCγ-d dendrite. We designated these as *en passant* synapses.

To estimate the prevalence of each synapse type we designed an automated method to roughly classify the connections between individual pairs of presynaptic visual inputs and downstream KCγ-d cells as either bouton-claw or *en passant*. For each individual pair of synaptic partners, we computed the centroid of synapse locations as well as the variance in synapse distance from this centroid. We predicted that bouton-claw synapses would have a lower variance in synapse distance from the computed centroid, while *en passant* synapses would have a higher variance in synaptic distance from a centroid. We manually determined the synapse type identities for a set of ground truth presynaptic VPNs/LVINs and postsynaptic KCs and used this ground truth to set an appropriate threshold for classifying synapse type using the variance in synapse distances from the centroid of all synapse locations. We then used a separate test set of 30 presynaptic-postsynaptic neuron pairs and manually identified synapse types for these connections to determine our method's accuracy; we found that 86% of connections could be reliably predicted in this test set. After validating this method, we determined the types for all synapses between VPNs or LVINs and their postsynaptic KCγ-d partners.

### Morphological cell type clustering of neurons

All VPN and LVIN inputs in both hemispheres were taken into account during the clustering process. We used hierarchical clustering to group

together neurons based on raw scores obtained from an all-by-all NBLAST run on all neuron skeletons. We then tuned clusters by eye by comparing morphologies and in some cases examining inputs, outputs, and hemilineages.

There are 6 neurons annotated as aMe26 in FlyWire (3 per hemisphere) but the neurons on left and right sides of the brain differ considerably in their morphology. Despite FlyWire calling these cells the same type, we chose to treat them as unique cell types based on our NBLAST clustering, and refer to these as aMe26_L and aMe26_R throughout the text and figures.

### Comparing visual input organization to KCγ-d's between left and right hemispheres

Left-Right neuron morphology matches for direct VPNs and LVINs were identified in a stepwise manner. For some neurons in Codex, there is an assigned "mirror twin," or neuron that is the morphological match in the opposite hemisphere. Whenever this was the case, we chose this neuron to be the match. If there was no mirror twin listed for a neuron, we searched for all neurons that shared the same Hemibrain Type and assigned matching left-right pairs by eye. As a final step, if the neuron did not have a match through Codex we searched for a match among our list of neurons connected to KCγ-d's. For neurons where we could not easily query other cells of potentially similar morphology, we designated the match as "Unknown". For neurons that had a morphological match, we queried whether the match was also directly upstream of any KCγ-d's in the opposing hemisphere using a ≥ 5 synapse threshold. Upon doing so, we found that 45 neurons were unique to either the left or right mushroom body circuits. If we removed the synapse threshold, we were able to find the match for 15 of these 45 neurons in the opposing hemisphere. Following the process described above, we obtained 124 neurons with direct inputs to KCγ-d's across both hemispheres with putative matches in the opposite hemisphere. Since mirror twins were more ambiguous in the case of neurons within the same type class with very similar morphologies (e.g., aMe12), we condensed the matrix by cell type in some cases by averaging connectivity across direct VPNs and LVINs of the same morphological class.

To measure the extent of structure or randomness to the connectivity of direct VPNs and LVINs onto KCγ-d's between hemispheres, we first counted the pairs of KCγ-d's across hemispheres that received the same pattern of direct VPN and LVIN input (similar to analysis done in Eichler et al.[13]). Specifically, we counted how many KCγ-d's in the left hemisphere had an identical pattern of input to a KCγ-d in the right hemisphere; the total number of possible matches was 147, the number of KCγ-d's in the left hemisphere. Matches between hemispheres involving KCγ-d's receiving 0 inputs from our subsets of left-right matched VPNs and LVINs were not counted. We then compared the number of such matches we observed in the FlyWire dataset to the number of matches observed after shuffling both the left and right connectivity matrices while preserving VPN and LVIN connection probabilities and the number of inputs to each KCγ-d (1000 shuffled models were used; see Random models of VPN/LVIN-KC connectivity).

We then used PCA to recover structure in direct VPN and LVIN connectivity to KCγ-d's after concatenating left and right connectivity matrices. We compared this structure to that obtained with random and stereotyped models of connectivity. For the random model, we shuffled connectivity across both hemispheres. In the stereotyped model, connectivity was shuffled in a randomly selected hemisphere and copied to the remaining hemisphere to generate identical but random connectivity in each hemisphere. In both models, shuffling was done to preserve input connectivity distribution and the number of inputs sampled by each KCγ-d (see Random models of VPN/LVIN-KC connectivity).

### Estimating receptive fields of direct and indirect VPNs

Only visual projection neurons with optic lobe processes in the left hemisphere were considered for receptive field analysis. When computing receptive fields for all VPNs (as in Fig. 4E, F), VPNs with inputs in the left optic lobe with receptive fields encompassing more than 5 ommatidial columns were included. Columnar markers (759 in total) as identified in[57] were used. Since most medulla VPNs considered in this study arborize between layers M6 and M8, we used M5 column markers that were previously identified through the reconstruction of Mi1 processes in layer M5. Lobular column markers were obtained by extension of the medulla column map to the lobula from interpolating the positions of 63 reconstructed TmY5a neurons, as described in Zhao et al.[57]. Each columnar marker was mapped to an ommatidial viewing angle, directional vectors that were represented by points on a unit sphere (the "eyemap").

To estimate the receptive field of each VPN, we used morphological information available in the FlyWire dataset to compute dendritic proximity to ommatidial columns within a specific distance threshold. We then computed the convex hull that encompassed this subset of ommatidial columns. A suitable threshold was set separately for lobular and medullar neurons by annotating receptive field edges by hand (visually selecting ommatidial columns) for a subset of ground truth neurons and choosing the threshold that maximized overlap between hand-drawn receptive fields and estimated receptive fields.

For all figures depicting receptive fields, a 2D Mollweide projection was used to portray the viewing directions of the ommatidial columns comprising the VPN's estimated receptive field. These viewing directions, as well as midline and equatorial lines in visual space, were determined in[57].

### Matching neurons between FlyWire and Hemibrain volumes

To establish a mapping between FlyWire VPNs/LVINs and their counterparts in the hemibrain dataset, we used the fafbseg-py package (https://github.com/navis-org/fafbseg-py) to fetch and process FlyWire meshes for all neurons with cell bodies in the left hemisphere. We then used the transforms and brain templates available in navis-flybrains[109] to transform these FlyWire meshes to the hemibrain brain template (specifically the JRC2018F template). As the FAFB brain was inverted during image acquisition, this transform aligned FlyWire neurons from the left hemisphere with hemibrain neurons from the right hemisphere.

NBLAST[64] was used to query our subset of FlyWire neurons against all available neurons in the hemibrain dataset. The mean of the forward and reverse score was considered to obtain raw scores for each query-target pair. Each FlyWire neuron was paired with its highest scoring hemibrain match; in cases where two FlyWire neurons paired with the hemibrain neuron, the match with the stronger score was retained and the remaining neuron was paired with its next best NBLAST match. This process was repeated until a one-to-one mapping was obtained.

To probe if VPN/LVIN to KCγ-d connectivity was random between the two connectomes, we used PCA to extract structure in an aggregated connectivity matrix containing direct VPN/LVIN to KCγ-d connectivity from one hemisphere of FlyWire and one hemisphere of hemibrain. Using a similar approach to the analysis comparing left and right hemisphere connectivity (see Comparing visual input organization to KCγ-d's between left and right hemispheres), we compared the observed structure with a random model shuffling connectivity across both connectomes. In the stereotyped model, connectivity was shuffled in one of the connectomes and copied to the remaining hemisphere to generate identical but random connectivity in each hemisphere.

### Random models of VPN/LVIN-KC connectivity

To determine if there was any structure in connectivity that wasn't accounted for by the non-uniform distribution of input connection probabilities or the non-uniform number of inputs to each output neuron, we used a shuffle procedure where the connection probabilities of inputs and the number of inputs to each postsynaptic output were fixed. PCA was used to extract correlations in connectivity matrices such as VPN connectivity to LVINs or direct VPN/LVIN connectivity to KCγ-d's. The fractions of variance explained by the top components were compared with those obtained with a set of randomly generated connectivity matrices using this shuffle procedure[28]. For each figure exploring the extent of random connectivity in this study, data was compared to 1,000 random shuffles.

### Estimating dimensionality of VPN/LVIN-KC connectivity

To quantify the observation that inputs to visual KCγ-d's are more skewed toward the top few input types than in the case of olfactory input to KCγ-m connectivity (Supplementary Fig. 5D), we predicted the dimension of visual input and olfactory input connectivity to KCs by computing the participation ratio defined in Gao et al.[62]:

$$PR\left(C^{Neuron}\right) = \frac{(\Sigma^{M}_{i=1}\mu^{i})^2}{\Sigma^{M}_{i=1}\mu^{i^2}} = \frac{[TrC^{Neuron}]^2}{Tr[C^{Neuron}]^2} \tag{1}$$

where $C^{Neuron}$ is the covariance matrix of visual and olfactory input connectivity to KCγ-d and KCγ-m populations respectively and $\mu_1, \mu_2, \dots \mu_M$ are the eigenvalues of these covariance matrices.

### Conditional input analysis

We implemented the conditional input analysis methods described by Zheng et al.[59]. Specifically, we counted the number of connections to the KC cell population from one input A given input from another input B for all possible pairs of input (each connection to a KC was counted as one connection regardless of synaptic count). We then generated 1000 random models of shuffled connectivity (see shuffle procedure described in *Random models of VPN/LVIN-KC connectivity*) to generate a null distribution of conditional input counts. For each pair of inputs, we then computed a z-score from this null distribution for the observed connectivity (the resultant "conditional input matrix") to determine if observed connectivity from a specific input was significantly more or less dependent on connectivity from a second input than would be expected in a random model of connectivity. To extract groupings of inputs based on these conditional input scores, we then applied unsupervised K-means clustering[61] to the conditional input matrix. Since the number of clusters to form through KMeans clustering was a free parameter, we repeated experiments over a range of reasonable cluster numbers (from 2 clusters to 10 clusters).

### Clustering KCγ-d's using connectivity

Spectral clustering was applied to binarized direct VPN and LVIN to KCγ-d connectivity in an attempt to group KCγ-d's into clusters. The optimal number of clusters (4 clusters) was selected by computing the silhouette score as a measure of clustering accuracy over a range of cluster numbers and selecting the number of clusters with the best score.

### Data visualization

Images of neurons and brain anatomy were generated through Codex using the 3D viewer. The CNS brain mesh used throughout the figures are from the codex layer "brain_mesh_v3." The brain mesh was originally generated by Schlegel et al.[47]. The neuropil volumes highlighted in Fig. 2 were obtained through Codex (Explore > Neuropils) and were originally generated by Dorkenwald et al.[48] by bridging light level meshes from the Drosophila JFRC2010 template brain[110] into FlyWire.

Heatmaps in Supplementary Fig. 5 were made using the fafbseg R package from natverse[109] (R version 4.3.1 (2023-06-16), RStudio Version 2023.06.1 + 524). Adjacency matrices were first generated for groups of input and output neurons using the flywire_adjacency_matrix() function. The resulting connectivity matrices were then binarized so that every neuron-neuron contact with ≥5 synapses was set to 1, and any contact of 0-4 synapses was set to 0. The binarized matrices were plotted using the standard R heatmap() function, which also hierarchically clustered the rows and columns based on the similarity of their synaptic partners. As inputs, we used neuroglancer scenes from the v630 production dataset (ngl.flywire.ai) depicting left hemisphere olfactory receptor neurons with input to the left antennal lobe; left hemisphere uniglomerular PNs that receive input from left hemisphere ORNs in the left antennal lobe; left hemisphere KCγ-m's with input from the left hemisphere uniglomerular PNs; LVINs and VPNs that provide direct input to the left hemisphere KCγ-d's; and KCγ-d's in the left hemisphere.

Remaining plots were made using a combination of Google Sheets and standard Python plotting packages (Matplotlib, Seaborn).

### Reporting summary

Further information on research design is available in the Nature Portfolio Reporting Summary linked to this article.

## Data availability

Data that support the findings of this study are included within this paper and its Source Data files. The data can also be accessed through the online FlyWire Connectome Data Explorer (https://codex.flywire.ai/) and online Hemibrain Neuprint Data Explorer (https://neuprint.janelia.org/). Source data are provided in this paper.

## Code availability

Neurons in the FlyWire connectome (v630) were queried using a combination of open source software and online packages - the natverse R package (R version 4.3.1 (2023-06-16), RStudio Version 2023.06.1 + 524) and the accompanying fafbseg library; the NAVis python library version 1.3.1 (https://navis.readthedocs.io/en/latest/index.html) and accompanying fafbseg-py package version 1.13.0 (https://github.com/navis-org/fafbseg-py); the flybrains package version 0.2.6 (https://pypi.org/project/flybrains/) and the online Connectome Data Explorer (Codex, https://codex.flywire.ai/). Data was queried from the hemibrain dataset using the neuprint-python package version 0.4.25 (https://connectome-neuprint.github.io/neuprint-python/docs/). Data were analyzed using a combination of open source software packages - the natverse R package (R version 4.3.1 (2023-06-16), RStudio Version 2023.06.1 + 524) and the accompanying fafbseg library; Google Sheets basic plotting functions; the NAVis python library version 1.3.1 (https://navis.readthedocs.io/en/latest/index.html) and accompanying fafbseg-py package version 1.13.0 (https://github.com/navis-org/fafbseg-py); seaborn version 0.11.2 (https://seaborn.pydata.org/index.html). Quantitative analyzes were conducted using scipy version 1.9.1 (https://scipy.org/), networkx version 3.1 (https://networkx.org/) and scikit-learn version 1.1.2 (https://scikit-learn.org/stable/), Receptive fields were plotted using the open source packages descartes version 1.1.0 (https://pypi.org/project/descartes/), alphashape version 1.3.1 (https://pypi.org/project/alphashape/), and geopandas version 0.12.2 (https://geopandas.org/en/stable/). Code used to access and analyze data are provided at the following GitHub repository: https://github.com/ishanigan/visual-inputs-to-mushroom-body.

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

## Acknowledgements

We thank the Princeton FlyWire team and members of the Murthy and Seung labs, as well as members of the Allen Institute for Brain Science, for the development and maintenance of FlyWire (supported by BRAIN Initiative grants MH117815 and NS126935 to Murthy and Seung). We also acknowledge members of the Princeton FlyWire team, Cambridge Connectomics Group, and the FlyWire consortium for neuron proofreading and annotation. Natverse R and Python packages[109] were relied on for querying connectivity and generating data visualizations. Development of the natverse including the fafbseg package was supported by the NIH BRAIN Initiative (grant 1RF1MH120679-01), NSF/MRC Neuronex2 (NSF 2014862/MC_EX_MR/T046279/1) and core funding from the Medical Research Council (MC_U105188491). Proofreading in Cambridge was supported by Wellcome Trust (203261/Z/16/Z) to G. Jefferis. We thank Alvaro Sanz-Diez for her help with the validation of the neural type clustering. Funding was provided by NIDCD R01DC018032, the Pew Charitable Trusts, the McKnight Endowment Fund, the Rita Allen Foundation, and the University of Michigan to E.J.C.; Hearing, Balance, and Chemical Senses Training Program (T32-DC000011) and HHMI Hanna Gray Fellowship to E.L.H.; DOE CSGF (DESC0022158) and the Gastby Charitable Foundation to I.G.; BRAIN R34NS128874, NIH R01EY029311, the Mathers Foundation, the Pew Charitable Trusts, the McKnight Endowment Fund, the Grossman Charitable Trust and the Kavli Foundation to R.B.; A.L.-K. was supported by the Burroughs Wellcome Foundation, the Gatsby Charitable Foundation, the McKnight Endowment Fund, and NIH award R01EB029858.

## Author contributions

I.G. and E.L.H. conceived of the study. I.G. and E.L.H. curated the list of visual projection neurons from FlyWire. I.G. did analyses for Supplementary Fig. 1D, Fig. 2D, F, G, Fig. 3E, F, Supplementary Fig. 3, Fig. 4, Supplementary Fig. 4, Fig. 5D–G, Supplementary Fig. 5D-G, Supplementary Fig. 6, Supplementary Fig. 7, Fig. 6C–E, and Fig. 7. E.L.H. did analyses for Fig. 1, Supplementary Fig. 1, Fig. 2B, C, E, Fig. 3A–D, Supplementary Fig. 2, Fig. 5B, C, Supplementary Fig. 5A–C, and Fig. 6A, B. A.L.K., E.J.C., and R.B. supervised the study. All authors contributed to writing the initial draft of the manuscript. The manuscript was read and approved by all authors. I.G. and E.L.H. are co-first authors; the order of co-first authors was determined by a coin toss. R.B. and E.J.C. are co-corresponding authors and contributed equally to this work.

## Competing interests

The authors declare no competing interests.
