## [Peer Review File · Nature Communications]

Diversity of visual inputs to Kenyon cells of the *Drosophila* mushroom bodyEditorial Note: This manuscript has been previously reviewed at another journal that is not operating a transparent peer review scheme. This document only contains reviewer comments and rebuttal letters for versions considered at *Nature Communications*.

REVIEWERS' COMMENTS

Reviewer #1 (Remarks to the Author):

(Reviewer #1 in the original review)

The authors have nicely addressed the points in my original review and I have no further major comments.

Minor points (I do not need to see it again)

1. The participation ratio should be defined mathematically in the methods so that readers don't need to track down the original reference. (In particular, I might not have made my original comment if I had understood that the participation ratio was exactly what I was looking for.)
2. Should Fig 5 Supplement 1d actually be Fig 6 Supplement 1 (because it relates to Fig 6c)? – I leave that for the editors to decide
3. Fig 5 Supplement 2 – on the color bar for the z-scores, I think there are missing negative signs on the blue colors.

Reviewer #2 (Remarks to the Author):

I thank the authors for their careful consideration of my previous comments. The new version addressed all the points raised in a consistent way. I had not understood (due to the scarcity of information) that the analysis of neurotransmitter identities had not been done by the authors; this is clear now and the procedure used by the team that did it also better explained.

Overall, I like very much this revision and the work performed.

Response to Reviewers

Thank you again to the reviewers for their constructive feedback on our manuscript. Below we address the 3 minor points raised by Reviewer #1.

Reviewer #1 (Remarks to the Author):

(Reviewer #1 in the original review)

The authors have nicely addressed the points in my original review and I have no further major comments.

Minor points (I do not need to see it again)

1. The participation ratio should be defined mathematically in the methods so that readers don't need to track down the original reference. (In particular, I might not have made my original comment if I had understood that the participation ratio was exactly what I was looking for.)

The participation ratio has now been defined in the Methods section.

2. Should Fig 5 Supplement 1d actually be Fig 6 Supplement 1 (because it relates to Fig 6c)? – I leave that for the editors to decide

We have now moved this panel to Figure 6.

3. Fig 5 Supplement 2 – on the color bar for the z-scores, I think there are missing negative signs on the blue colors.

Thank you for catching this. The negative signs have been restored.

Reviewer #2 (Remarks to the Author):

I thank the authors for their careful consideration of my previous comments. The new version addressed all the points raised in a consistent way. I had not understood (due to the scarcity of information) that the analysis of neurotransmitter identities had not been done by the authors; this is clear now and the procedure used by the team that did it also better explained.

Overall, I like very much this revision and the work performed.

Thank you!